# Significance of Midkine Signaling in Women’s Cancers: Novel Biomarker and Therapeutic Target

**DOI:** 10.3390/ijms26104809

**Published:** 2025-05-17

**Authors:** Emily J. Aller, Hareesh B. Nair, Ratna K. Vadlamudi, Suryavathi Viswanadhapalli

**Affiliations:** 1Department of Obstetrics and Gynecology, University of Texas Health San Antonio, San Antonio, TX 78229, USA; aller@livemail.uthscsa.edu (E.J.A.); nairh@uthscsa.edu (H.B.N.); 2Mays Cancer Canter, University of Texas Health San Antonio, San Antonio, TX 78229, USA; 3Audie L. Murphy Division, South Texas Veterans Health Care System, San Antonio, TX 78229, USA

**Keywords:** MDK, ovarian cancer, endometrial cancer, breast cancer, biomarker, women’s cancer, tumor microenvironment

## Abstract

Midkine (MDK) is a multifunctional protein that is secreted into the extracellular space. It functions as a cytokine or growth factor, modulating a variety of signaling pathways implicated in angiogenesis, antitumor immunity, metastasis, and therapy resistance. MDK overexpression has been documented in a variety of cancers, including those that affect women. MDK mediates its effects through activation of key signaling pathways such as MAPK/ERK, PI3K/AKT, and STAT3, which are pivotal for cell cycle progression, survival, and maintenance of stemness. Obesity and estrogen signaling, a known critical driver of women’s cancer, further elevate the levels of MDK. MDK’s effects are mediated by a variety of membrane receptors, such as integrins, protein tyrosine phosphatase ζ (PTPζ), anaplastic lymphoma kinase (ALK), and neurogenic locus notch homolog protein 2 (Notch2). Recently published studies have indicated that MDK is a potential therapeutic target and a biomarker for the progression of women’s cancer. In this review, we have provided a concise summary of the most recent papers that have examined the potential biomarker and therapeutic utility of MDK signaling in women’s cancer.

## 1. Introduction

Midkine (often abbreviated MDK or MK) is a small, heparin-binding protein. It was first discovered in 1988 as a gene highly expressed in mouse embryogenesis during the midgestation phase [1,2]. While MDK is expressed during gestation, where it plays a role in neurogenesis [3], its production dramatically decreases prior to birth. MDK generally has low expression in the adult human body, though some tissues including the kidneys, bronchial epithelium, epidermis [4], ovaries [5,6], adipocytes [7], and endometrium [8] do express the protein under physiological conditions. In contrast, MDK is highly expressed in several pathological conditions, including autoimmunity [9,10,11], dyslipidemia [12], atherosclerosis [12,13], obesity [14], and most important for this review, many cancers [15]. In a malignant context, several cell types have been found to produce and secrete MDK, including tumor cells [16], cancer-associated fibroblasts [17], and immune cells [18]. Building on the body of literature characterizing MDK as a cancer-promoting protein, this review article will discuss the recent literature with a focus on the role and therapeutic implications of targeting the MDK axis to treat women’s cancers.

## 2. Midkine Structure and Regulation

MDK is a 15.5 kDa secreted protein that functions as a growth factor and cytokine. It is encoded by the *MDK* gene, which consists of four coding exons [19] and is located on the 11q11.2 chromosome [20]. Alternative splicing of the MDK mRNA can produce at least seven different isoforms of the mRNA [4]. There exist at least five different isoforms of mature protein, including the conventional MDK, a slightly extended form known as VA-MDK, and various truncated (tMDK) forms resulting from the skipping of entire exons or portions thereof (Figure 1) [4,21]. Notably, while several of the tMDK forms are found in neoplasms, VA-MDK can be found in significant quantities alongside conventional MDK in human tissues, though its biological significance remains to be clarified [4,22]. MDK is a member of the midkine family of proteins, which includes two heparin-binding growth factors: midkine and the structurally similar protein pleiotrophin [23]. MDK protein structure is conserved between mammals, rodents, fish, and *Xenopus*, and is enriched in cysteine and basic residues such as arginine and lysine [24]. The MDK protein consists of an N domain and a C domain connected via a hinge region [22]. Of the two, the N domain seems to be important for stability and is involved in MDK dimerization, while the C domain may be more important in MDK function [22]. Within the C domain, there are two heparin binding sites. Additionally, a short, 20aa signal peptide precedes the N domain of the main protein chain, targeting MDK for secretion [4,22].

MDK transcription and translation can be regulated through a number of factors. Binding sites for retinoic acid receptors [19,25], estrogen receptors [26], hypoxia-inducible factor 1α (HIF-1α) [27], the Wilms tumor suppressor gene [28], specificity protein 1 [29], and p53 [30] have been identified in the MDK promoter region, with each implicated in promoting MDK expression. Conversely, several microRNAs (miRNAs) have been demonstrated to reduce MDK translation, including miR-9 [31], miR-124 [32], miR-624 [33], miR-1275 [34], and miR-188 [35].

## 3. Midkine Signaling Pathways

### 3.1. Midkine Receptors

Midkine is a pleiotropic cytokine that acts through binding to a sizable repertoire of receptors, including both proteoglycans and non-proteoglycans. Proteoglycans with which MDK binds include protein tyrosine phosphatase ζ (PTPζ) [36,37], syndecans [38], and glypicans [39,40]. In addition to proteoglycans, several non-proteoglycans serve as key receptors of MDK signaling, including anaplastic lymphoma kinase (ALK) [41], low-density lipoprotein receptor-related protein 1 (LRP1) [42], neurogenic locus notch homolog protein 2 (Notch2) [43], integrins [44], and sortilin-related receptor 1 (SORL1) [45,46] (Table 1). While PTPζ, α4β1 integrins and α6β1 integrins have been demonstrated to form a MDK receptor complex [44], and some evidence suggests that both ALK and neuroglycan-2 can be components of MDK signaling complexes [22], it remains unclear if forming a complex is required for all MDK receptors (Table 1).

While most of MDK’s activity seems to occur in response to extracellular MDK binding to cell surface receptors, there is evidence that MDK may have intracellular functionality as well. LRP1 can promote MDK binding to the nucleocytoplasmic shuttle protein nucleolin (NCL) localized to the cell surface [47]. This then induces endocytosis, which seems to facilitate some of MDK’s immunomodulatory roles and promote cell survival [48,49]. Additionally, a recent study has demonstrated that intracellular MDK interacts with serine/threonine kinase 11 (LKB1) and STE20-related adaptor alpha (STRAD) to ultimately reduce LKB1 activity and decrease AMP-activated protein kinase (AMPKα) phosphorylation [50].

Midkine activates multiple cell-signaling pathways, with one of the pathways most commonly dysregulated pathways in cancers, including women’s cancers, being particularly affected [51]. The mammalian target of rapamycin (mTOR) pathway, a downstream target of PI3-kinase (PI3K)/protein kinase b (AKT), is also activated by MDK. In melanoma cells MDK activates mTOR to promote lymphangiogenesis [52]. Another major MDK pathway is the MAP-kinase (MAPK)/extracellular signal-regulated kinase (ERK) pathway, which is involved in growth and proliferation [53]. MDK has also been shown to activate nuclear factor kappa-light-chain-enhancer of activated B cells (NF-κB) signaling [54] and signal transducer and activator of transcription 3 (STAT3) signaling [7,55], and it can increase the expression of yes-associated protein (YAP) and transcriptional coactivator with PDZ-binding motif (TAZ) of the Hippo/Yap pathway in lung cancer [35]. Furthermore, MDK has been linked to the wingless and integrated (Wnt)/β-catenin pathway, with some studies demonstrating an activating role for MDK [56] and others demonstrating an inhibitory role [57]. In non-small-cell lung cancer, MDK derived from cancer-associated fibroblasts is capable of promoting radioresistance by enhancing glycolysis via Wnt/β-catenin activation [56]. In contrast, MDK would seem to suppress Wnt/β-catenin signaling in osteoblasts via binding PTPζ [57]. This would suggest that MDK may be capable of exerting differing effects on its downstream pathways depending on the signaling context.

**Table 1 ijms-26-04809-t001:** MDK effects on cancer cells.

MDK Receptor	Known Downstream Pathways	Function	References
ALK	PI3k/AKT, MAPK/ERK, KRAS/RAF	Promotes cancer cell growth and proliferation; suppresses apoptosis; promotes angiogenesis	[41,54]
Glypicans	PI3K/AKT	Promotes proliferation, adhesion, migration and invasion	[39,40]
Integrins	LRP6 and PTPζ	Facilitates cancer cell migration; promotes T-cell exhaustion	[44]
LRP1	LKB1/AMPKα, PI3K/AKT	Enhances proliferation of cancer cells; promotes recruitment and M2 polarization of tumor-associated macrophages; promotes adhesion and extravasation of PMNs; promotes T-cell exhaustion. Facilitates angiogenesis and anchorage independent growth	[37,42,58,59],
Notch2	JAK2/STAT3	Promotes EMT, proliferation, and growth of cancer cells; promotes M2 polarization of tumor-associated macrophages	[43]
NCL	LKB1/AMPKα	Promotes cancer cell proliferation; important in crosstalk between cancer cells and the tumor microenvironment, promotes a malignant phenotype in endothelial cells and and helps to promote an immunosuppressed tumor microenvironment.	[47,48,49]
PTPζ	KRAS/RAF; PI3K/AKT/mTOR, NFκB	Promotes cancer cell proliferation and suppresses apoptosis	[36,37,57]
SORL1		Promotes M2 polarization of tumor-associated macrophages	[59]
Syndecans	PI3K/AKT, MAPK/p38	Promotes Treg recruitment, T-cell exhaustion, M2 polarization of tumor-associated macrophages, and lymphangiogenesis	[38,60,61]

MDK signaling is implicated in diverse functions, such as mitogenicity, inflammation, angiogenesis, metastases, and stem cell self-renewal [62]. The effects of MDK are determined in a tissue-specific manner by the MDK receptors present in cells. Membrane receptors, such as integrins [44], PTPζ [62], ALK [60], and Notch2 receptor [43], are reported to serve as MDK receptors in cancer cells. Further MDK also interacts with LRP1 [42] and syndecans [60] in other tissues.

### 3.2. Midkine in Cell Proliferation

MDK is implicated as a critical factor in cancers by promoting cell proliferation, suppressing cell death, and ultimately promoting tumor growth. MDK’s main signaling pathway, PI3K/AKT, is a key regulator of cellular survival and proliferation [63] (Figure 2A). Correspondingly, MDK has been repeatedly linked to enhanced proliferation in various malignant and non-malignant contexts [29,50,58,64]. MDK can promote proliferation of various members of the tumor environment, including endothelial cells [41] and cancer-associated fibroblasts [65]. The ability of MDK to suppress cell death is another likely contributing factor to the cytokine’s ability to promote tumor growth (Figure 2). MDK expression can induce the expression of pro-survival signals such as B-cell lymphoma 2 protein (BCL2), cyclin D1, and ERK [43]. Enhancement of the expression of BCL2 has been repeatedly identified as a mechanism by which MDK suppresses cell death, and this expression can be maintained even when challenged with an apoptosis-inducing agent like cisplatin [66].

Consistent with MDK’s ability to suppress cell death, evidence suggests that it also plays a role in preventing cell death in response to cancer therapies. MDK can reduce caspase-3 cleavage in response to doxorubicin treatment [67]. Exogenous MDK and MDK from cancer-associated fibroblasts was able to decrease cisplatin-induced cell death in oral squamous cell carcinoma, ovarian cancer, and lung cancer by upregulating the expression of BCL2 [17]. Additionally, MDK from cancer-associated fibroblasts upregulated multidrug resistance protein 1 and multidrug resistance protein 2 to promote export of cisplatin from the cell [17]. Cancer-associated fibroblast-secreted MDK has also been implicated in facilitating cellular survival after radiation therapy by promoting glycolysis and DNA repair via activation of the Wnt/β-catenin pathway [56].

### 3.3. Midkine in Estrogen Signaling

While MDK is elevated in most cancers, its link to estrogen signaling presents a notable connection between MDK and women’s cancers. Estrogen signaling plays a role in several women’s cancers, including breast [68], endometrial [69], and ovarian cancers [70,71]. MDK, having an estrogen response element in its promoter [19], has been repeatedly demonstrated to be induced by estrogen signaling [19]. In normal human ovarian follicles, MDK expression correlates with estradiol concentration [5], and in normal endometrial cells and ovarian fibroblasts estradiol stimulates MDK expression [72,73]. Dienogest, an artificial progestin that decreases circulating estradiol and estrogen production [74], also decreases MDK secretion [75]. Similarly, MDK is highly expressed in endometrial tissue during the proliferation phase, which is marked by high estrogen signaling [8]. Ovarian estrogen secretion promotes ovarian metastasis of gastric cancers by enhancing the secretion of MDK by fibroblasts [73]. The ability of estrogen to promote MDK signaling is maintained in cancer cells as well, with *MDK* being an estrogen-induced gene in both lung [26] and endometrial cancer cells [76].

Obesity is known to promote women’s cancers through the enhanced secretion of estrogen as well as through other cytokines [77,78]. Interestingly, a set of studies has indicated that MDK may be involved in or promoted by obesity. One study found that human adipocytes express MDK and that MDK is elevated in the tissue of obese mice and serum of obese humans [14]. Plasma MDK levels were found to correlate with body mass index (BMI) in humans [14]. Furthermore, in human adipocytes MDK activated the STAT3 pathway [14]. Another study has demonstrated that MDK can be secreted by adipose progenitor cells, which are expanded in obesity, to recruit monocytes to the resident tissue [79]. MDK was found to be correlated with waist circumference, triglycerides, and total cholesterol level and negatively correlated with high-density lipoprotein in patients with hidradenitis suppurativa [80]. However, one report did not find any correlation between BMI and serum MDK levels in adults or children [81], challenging the previous reports. Nevertheless, a recent study supports a possible role for MDK in obesity-driven cancers, as MDK was identified as an upstream regulator involved in the gene expression changes and DNA damage accumulation seen in breast tissues in obese (BMI > 25) and non-obese (BMI 20–25) women with *BRCA1* and *BRCA2* mutations [82]. These emerging studies implicate estrogen as a promoter of MDK signaling, but how and if this translates to obese patients with elevated estrogen remains unclear.

### 3.4. Midkine in Cancer Stemness

MDK signaling is implicated in the self-renewal and tumorigenic potential of stem cells [83]. MDK enhances the growth and survival of normal stem cells, including both embryonic stem cells [84] and mesenchymal stem cells [85], in hypoxic conditions [85]. In both murine embryonic stem cells and human embryonic stem cells, MDK enhances self-renewal by promoting Nanog and octamer-binding transcription factor 4 expression via the PI3K pathway [86]. A MDK-PI3K/AKT axis for modulating stem cells was also identified in breast cancer [34], and activation of Hippo/YAP levels via suppression of YAP/TAZ levels has also been implicated in MDK-mediated enhancement of cancer stem cell activity *in vitro* and *in vivo* [35].

Cancer stem cells are major mediators of therapy resistance, and MDK-mediated enhancement of cancer stem cells has been linked with improved cancer stem cell survival following epirubicin treatment [34]. Furthermore, inhibition or knockdown of MDK suppressed the growth and migration of prostate cancer stem cells, enhancing the efficacy of docetaxel treatment [87]. Taken together, the results from these studies suggest that MDK may be involved in promoting drug resistance via enhancing persistence of the cancer stem cell population in cancers exposed to chemotherapies.

### 3.5. Midkine in Angiogeneiss

Midkine is a proangiogenic factor (Figure 2B). As an important inducer of endothelial cell proliferation, MDK has been linked to angiogenesis and arteriogenesis in a non-malignant context [88]. MDK is secreted by human umbilical vein endothelial cells (HUVECs) under conditions of hypoxia and is capable of promoting neovascularization in chick chorioallantoic membrane (CAM) models [89]. MDK^−/−^ mice exhibited impaired revascularization following hind limb ischemia [89]. Further evidence indicates that both endothelial cells and immune cells (polymorphonuclear cells (PMNs) and monocytes) can produce MDK to promote endothelial cell proliferation [89,90].

In the context of malignancy, there is substantial evidence for MDK promoting tumor growth and survival by enhancing access to vasculature. Midkine has been repeatedly shown to enhance the vascular density and endothelial cell proliferation of tumors across various cancers [91,92,93]. MDK derived from breast [91] and liver [92] cancer cells enhanced angiogenesis in a rabbit corneal assay and a CAM model, respectively. In HUVECs, overexpression of MDK enhances migration and neovascularization, whereas inhibition via either small interfering RNA (siRNA), an MDK-binding peptide, or an inhibitor reduced HUVEC proliferation [94], migration [93], and tube formation [95].

While there are some slight variations in the pathways by which MDK activates endothelial cell proliferation and angiogenesis, AKT serves as a common denominator. Stoica et al. have demonstrated that MDK binds to ALK and activates AKT and MAPK phosphorylation to promote endothelial cell proliferation [41]. Similarly, miR-9 was found to inhibit angiogenesis in nasopharyngeal carcinoma by inhibiting MDK and reducing signaling via a pyruvate dehydrogenase kinase 1/AKT/p70S6 kinase pathway [31]. The role of another major angiogenic factor, vascular endothelial growth factor (VEGF), is less consistent. The deubiquitinase ubiquitin-specific protease 12 (USP12) can promote angiogenesis in breast cancers by stabilizing MDK and promoting the downstream activation of the AKT pathway and VEGF receptor 3 (VEGFR3) [96]. Another study found that MDK^−/−^ mice had reduced plasma bioavailability of vascular endothelial growth factor A, which inhibited revascularization following ischemia in a mouse hind limb ischemia model [90]. Conversely, while MDK overexpression was found to promote enhanced angiogenesis in a CAM experiment, MDK knockdown failed to reduce the level of VEGF, suggesting that MDK does not regulate VEGF and can promote angiogenesis independently of VEGF [93]. It is possible that these signaling inconsistencies may be due to variability in the receptors with which MDK binds—as there is evidence that MDK can exert different vascular effects depending on which integrin(s) it binds to, α4 or α6 [97].

### 3.6. Midkine Signaling in Hypoxia

Hypoxic conditions are common in cancers, and they are correlated with a poor prognosis in endometrial [98], breast [99], ovarian [100], and cervical [101] cancers. As a pro-angiogenic factor, MDK can play a key role in mitigating hypoxic death. In murine embryonic stem cells, MDK binding to LRP1 protected cells from hypoxia-induced apoptosis via AKT, HIF-1α, and heme oxidase signaling [84]. Furthermore, mesenchymal stem cells overexpressing MDK were resistant to hypoxia-induced apoptosis and had enhanced survival following cardiac infarction in mice [85]. Similarly, the hypoxic conditions often found in cancers can also promote MDK signaling. As the HIF-1α binding site in MDK’s promoter would suggest, hypoxia promotes MDK expression in the lungs [27] of mice as well as human PMNs [89], monocytes [89], HUVECs [89], glioblastoma [102], hepatocellular carcinoma [103], and non-small-cell lung cancer [93]. In normal ovarian follicles, MDK also demonstrated a negative correlation with O_2_ concentration [5].

### 3.7. Midkine Signaling in Metastasis

MDK promotes metastasis in multiple ways, including by promoting epithelial to mesenchymal transition (EMT), facilitating migration and invasion, and helping to form the premetastatic niche. Of these, MDK’s role in EMT is best characterized, and MDK is capable of promoting both the proliferation of mesenchymal cells and the acquisition of mesenchymal characteristics [104]. Several of the pathways that MDK contributes to serve as key regulators of EMT, including PI3K/AKT/mTOR, NF-kB, and Wnt/β-catenin, and through these pathways MDK has been linked to EMT [105]. MDK secreted from cancer-associated fibroblasts promoted the expression of EMT genes in gastric cancer via the PI3K/AKT pathway, and through this conferred cisplatin resistance [106]. The interaction of glypican 2 (GPC2) and MDK promoted enhanced migration and invasion via the PI3K/AKT pathway as well, with MDK overexpression restoring the reduced migration and invasion in GPC2-depleted cells [39].

In glioblastoma, MDK was demonstrated to enhance EMT, migration, and invasion, presumably through the PI3K/AKT pathway [102,107]. MDK-Notch2 signaling has also been linked to its ability to promote EMT through the downstream activation of NF-kB, hairy and enhancer of split 1, and through crosstalk with the Janus kinase (JAK)-STAT3 pathway [43,93,108]. Another STAT family member, STAT1, was found to promote EMT by upregulating MDK in response to interferon gamma (IFN-γ) treatment [109]. Inhibiting MDK following treatment with IFN-γ reduced both the expression of EMT markers like Snail and vimentin and reduced the migration and invasion of various cancer lines. This would suggest a potential therapeutic benefit to adding MDK inhibitors to IFN-γ therapy (Figure 2A).

MDK has also been demonstrated to promote metastasis *in vivo.* In non-small-cell lung cancer, MDK knockdown abrogated lung metastases in vivo, while increasing MDK promoted a ninefold increase in metastatic burden [93]. In an orthotopic breast cancer model, silencing USP12—an MDK stabilizing protein—reduced lung metastases in a manner that could be reversed through MDK supplementation [96].

Aside from its role in promoting EMT and the acquisition of migratory capacity, MDK can also promote metastasis by modulating the premetastatic niche. Through promoting lymphangiogenesis via mTOR signaling, MDK can promote metastasis to both lymph nodes and visceral organs [52]. Conversely, reducing MDK expression resulted in a marked decrease in lymphangiogenesis and both lymph node and distant metastasis [52]. This would therefore suggest that MDK may be a promising target for limiting the metastatic spread of cancers, by simultaneously limiting the acquisition of mesenchymal phenotypes, migratory ability, and access to the lymphatic vasculature important for tumor spread.

### 3.8. Midkine Signaling in the Tumor Microenvironment

MDK is known to modulate the tumor microenvironment by influencing the survival [110] and recruitment [111,112] of immune cells, cytokine secretion [112,113], and angiogenesis [113]. Early studies found that MDK^−/−^ mice exhibited impaired leukocyte migration [9,114,115]. However, further investigations suggest that MDK plays various roles in immune regulation, and that its effects may differ depending on the context of disease. For example, MDK has been investigated as a mediator of autoimmune disease, where it was found to suppress regulatory T-cell (Treg) and regulatory dendritic cell expansion [11,116]. However, in the context of cancer, MDK generally promotes immunosuppression by regulating the various cell populations of immune cells, including Tregs (Figure 2C). Indeed, MDK expression facilitates enhanced Treg infiltration in several cancers [112,117]. MDK expressed by tumor cells can promote enhanced Treg infiltration by promoting Treg motility, with syndecan 4 (SDC4) serving as the putative receptor for MDK on Treg cells [61]. This interaction was seen even in early colorectal cancer cases, suggesting that MDK may play a role in establishing the tumor microenvironment [61]. In hepatocellular carcinoma, MDK was associated with enhanced Treg and myeloid-derived suppressor cell (MDSC) infiltration and enhanced expression of immune checkpoint molecules such as cytotoxic T-lymphocyte-associated protein 4 (CTLA4) [103]. Alongside this, MDK was associated with the expression of markers of immunosuppression including arginase-1 (Arg1), transforming growth factor beta (TGF-β), inducible nitric oxide synthase (NOS2), and interleukin 10, demonstrating that MDK is involved in facilitating a cancer-promoting tumor microenvironment [103]. Taken together, these studies highlight a key area of uncertainty regarding the mechanisms by which MDK is capable of promoting hyperimmunity in some contexts and immune suppression in others.

Another element of the immune microenvironment that is regulated by MDK is macrophages. Not only does MDK promote macrophage/monocyte migration, MDK has been repeatedly linked to promoting M2 polarization of macrophages into tumor-associated macrophages [59,117]. MDK can promote macrophage infiltration into melanoma tumors, and it promotes expression of TGF-β and interleukin 13—markers of immunosuppressed environments [112]. Histopathological analysis have demonstrated a correlation between MDK and tumor-associated macrophages, and loss-of-function and gain-of-function studies demonstrate an ability for MDK to enhance expression of tumor-associated macrophage markers, including Arg1, NOS2, and chitinase-like protein 3 [112]. This MDK-mediated macrophage polarization was found to occur in a NF-kB dependent manner [112]. This ability of MDK to promote M2 polarization and macrophage infiltration has been demonstrated in several cancers, with several putative receptors being identified including LRP1, Notch2, and syndecan 3 (SDC3) [59,117,118]. Furthermore, one study found that MDK can promote phagocytic activity and lipid metabolism in macrophages, conferring a phenotype that promotes osimertinib resistance in metastatic non-small-cell lung cancer [118]. This would suggest that MDK’s ability to influence the tumor microenvironment may support tumor survival beyond strictly reducing immune-mediated killing of cancer cells.

MDK has also been linked to the regulation of other members of the tumor immune environment. MDK is important for PMN adhesion and extravasation, which it promotes by binding to LRP1 and promoting the high affinity conformation of β2 integrins [119]. MDK inhibits the infiltration of type 1 and type 2 dendritic cells, and it can be secreted by monocytes, macrophages, and dendritic cells in response to stimulation by toll-like receptors 3 (monocytes), 4 (macrophages and some dendritic cell subtypes), and 7/8 (monocytes and some dendritic cell subtypes) [18,112,120,121]. In breast cancer, MDK treatment led to a reduction in natural killer (NK) cells [120]. MDK can also reduce the cytotoxicity of NK cells by enhancing the production of secreted MHC class 1 polypeptide-related sequence A and B (MICA/B) via a p38MAPK/ C/EBP homologous protein axis [122]. Elevated levels of secreted MICA/B bind to the natural killer group 2D receptor on NK cells and promotes endocytosis of the receptor, decreasing NK cell ability to bind and recognize MICA/B on tumor cells [122].

MDK’s role in modulating T-cell activity is less consistent. There exists some evidence that MDK can activate CD8^+^ T cells to secrete chemokine (C-C motif) ligand 4 (CCL4) in neurofibromatosis type 1 patients via interaction with LRP1 on T cells, but the effect of this interaction on cytotoxicity was not investigated [123]. In cancers, though, MDK seems to exert a primarily inhibitory effect on CD8^+^ T cells, as there is evidence that MDK can be secreted by tumor cells to suppress the antitumor activity of CD8^+^ T cells [112,124]. MDK’s suppressive effects on T cells may occur indirectly through the activity of tumor-associated macrophages, MDSCs, or other immunosuppressive cells rather than direct action of MDK [103,112]. Indeed, another study found that MDK treatment increased the expression of TGF-β and hepatocyte growth factor by macrophages, both of which induce T-cell exhaustion and Treg activity [118]. However, MDK receptors do exist on the surface of T cells, and they are correlated with a decreased sensitivity to programmed cell-death protein 1 (PD-1) inhibitors suggesting that MDK may be able to exert direct effects as well [124] (Figure 2C).

In a recent study published by Luo et al., MDK’s immunosuppressive ability was linked with the development of resistance to cyclin-dependent kinase 4/6 (CDK4/6) inhibitors in metastatic breast cancer [125]. MDK-NCL and MDK–syndecan interactions between tumor cells, stromal cells, and immune cells were found to be increased in tumors with acquired resistance to CDK4/6 inhibitors, though it was not associated with intrinsic resistance and early progression [125]. As a whole, MDK’s apparent role in modulating the tumor immune microenvironment emphasizes its potential clinical utility.

## 4. Midkine Signaling in Women’s Cancers

### 4.1. Midkine Signaling in Breast Cancer

MDK has been extensively linked to breast cancer (Figure 3). Its gene expression is higher in breast cancer tissues compared to normal breast tissues [126,127]. Among the elevated MDK forms, both conventional and tMDK have been detected in breast cancer samples [128], although the functional significance of tMDK remains unclear. MDK overexpression is observed across all major breast cancer subtypes, including estrogen receptor (ER)-positive, human epidermal growth factor receptor 2 (HER2)-positive, and triple-negative breast cancer (TNBC) [129]. Elevated MDK levels have been associated with lymph node metastases and advanced TNM stage, but no significant correlation has been reported with age, tumor size, or menopausal status [129]. Consistent with these findings, immunohistochemistry analyses have shown that MDK protein levels correlate with the clinical stage, as well as T, N, and M classifications in breast cancer patients [127].

A recent study has identified MDK as a driver of age-related changes in breast tissue and a key component of a “MDK-age gene signature”, which can predict the prognosis of ER-positive breast cancer [120]. Furthermore, MDK treatment enhanced tumorigenesis in vivo and correlated with increased cancer cell proliferation in younger patients with ER-positive breast cancer, which the authors link to sterol regulatory element-binding transcription factor 1 (SREBF1) via PI3K/AKT/mTOR signaling [120]. Notably, this study also found that high MDK expression in the breast tissue of young women correlated with a higher Gail-5 score, a model for predicting breast cancer risk. This would suggest that MDK may play a role in early tumorigenesis of ER-positive breast cancer. NF-κB is also involved in MDK signaling in breast cancer, with MDK activating NF-κB and downstream *NR3C1* to promote proliferation and migration of tumor cells by facilitating EMT [130].

Regulation of the tumor microenvironment also appears to be linked to MDK’s role in the progression of breast cancer. In inflammatory breast cancer, CD151, through the CD151-α6β1 integrin complex, promoted the release of MDK associated with extracellular vessels to recruit CD68^+^ monocytes to malignant areas [111]. Interestingly in most cancers, including breast cancer, increased CD68^+^ macrophage infiltration is correlated with poorer outcomes, similar to the trend seen with MDK [131]. This may be due to the fact that CD68^+^ is a marker for macrophages in general, whereas the increased infiltration in MDK^+^ tumors may be predominantly tumor-promoting M2 polarized macrophages. In breast cancer with leptomeningeal metastases, circulating tumor cells in the cerebrospinal fluid (CSF) were found to have elevated MDK, which could interact with circulating macrophages and monocytes via the LRP1 or SORL1 receptors [132]. Similarly, while MDK expression was negligible in the immune cells of normal CSF, both macrophages and type 1 conventional dendritic cells exhibited enhanced MDK expression in the CSF of patients with leptomeningeal metastases [132]. Thus, MDK represents an appealing drug target to interfere with various pro-tumor processes in breast cancer.

### 4.2. Midkine as a Biomarker in Breast Cancer

MDK has been identified as a potential biomarker in breast cancer. MDK is elevated in the plasma of patients with breast cancer, relative to normal plasma [126,133,134]. Furthermore, the likelihood of elevated plasma MDK increases with more advanced disease [133]. In a comparative study, serum MDK was found to not only be elevated in breast cancer patients, but to decrease following the surgical removal of the tumor [134]. As a prognostic factor, MDK has been found to be negatively correlated with the disease-specific survival in ER-positive breast cancer patients below 55 years old [120]. Indeed some studies have evaluated MDK as a potential prognostic marker. One study found that MDK was better at predicting cancer than the conventional plasma markers (cancer antigen 15-3 (CA15-3), carcinoembryonic antigen (CEA), and NCCST-439 antigen) both alone and when combined [133]. Furthermore, MDK has been evaluated as a component in gene panels for predicting breast cancer prognosis. An immune-regulator gene signature including MDK (alongside *TSLP*, *BIRC5*, *S100B*, *S100P*, *RARRES3*, *BLNK*, and *ACO1*) was found to be an accurate predictor of progression-free survival in breast cancer at 3 and 5 years, even within clinical groups (i.e., ER-positive, ER-negative, node-positive, node-negative etc.) [126].

Although MDK levels have been identified as an independent marker of prognosis in breast cancer, being associated with metastasis and reduced disease-free survival [127], its utility as a diagnostic biomarker remains limited. While high MDK expression correlates with more aggressive disease, including metastatic potential, this does not necessarily translate to reliable detection of malignancy in early-stage tumors. For example, a pilot study evaluating MDK levels in fine needle aspiration samples found elevated MDK in malignant tumors relative to benign ones in only a small subset of cases, and whether this subset carries prognostic value has not been explored [135]. Thus, MDK may be more reflective of disease progression than initial diagnosis.

### 4.3. Midkine in Ovarian Cancer

MDK and its receptors have been demonstrated to be affiliated with ovarian cancer (Figure 3). Immunohistochemical analysis indicates that MDK was elevated in ovarian cancer, and serous ovarian cancer in particular [136,137]. Furthermore, it was markedly associated with a poor differentiation grade [136]. In addition to MDK itself, both the LRP1 and ALK receptors have been found to be associated with ovarian cancer. Cytoplasmic ALK overexpression in high-grade serous ovarian cancer is correlated with a poorer prognosis, enhanced cancer stem cell features, enhanced migration, and proliferation [138]. On the other hand, LRP1 promotes the migration of ovarian cancer and is associated with poorer survival [139]; as a part of a panel of nine genes, it can predict high-risk patients with poorer outcomes in ovarian cancer and other cancers [140]. MDK has been proposed to suppress apoptosis in ovarian cancer, as reducing MDK mRNA via miR-124 promoted apoptosis of SKOV3 cells [32]. MDK has also been linked to promoting IFN-γ-induced metastasis [141]. MDK is a downstream target of IFN-γ via STAT1 and can be induced by IFN-γ signaling in a dose-dependent manner. This, in turn promotes metastasis via enhancing proliferation, migration, invasion, and EMT [141]. MDK secreted from ovarian cancer epithelial cells can also regulate cancer-associated fibroblasts by binding to NCL/syndecan 2 (SDC2)/LRP1. Conversely, cancer-associated fibroblasts communicate with epithelial cancer cells via MDK targeting SDC2/SDC4/NCL on the epithelial cell surface [142].

Although MDK is generally linked to the augmentation of therapeutic resistance, in ovarian cancer, evidence has emerged that both corroborate and contradict this notion. Big-data analysis found that epigenetic silencing of MDK seemed to promote cisplatin resistance, although no mechanistic exploration or experimental confirmation was performed [143,144]. Supporting this, Wu et al. found that MDK expression was associated with better response to combined paclitaxel/cisplatin therapy. In this case, MDK promoted the intracellular accumulation of these drugs via inhibition of multidrug resistance protein 3 [136]. In contrast, Mirkin et al. found that cisplatin resistant ovarian cancer cells expressed more MDK than drug-naive counterparts [67]. It may be that MDK promotes therapy resistance in ovarian cancer via the tumor microenvironment. Cancer-associated fibroblast-derived midkine was found to reduce cell death following cisplatin treatment via increasing the long non-coding RNA lncANRIL [17]. Additionally, an MDK-NCL axis was found to promote therapy resistance via increasing CD8^+^ T-cell exhaustion, although this was most impactful before and during neoadjuvant chemotherapy and less important after [145].

### 4.4. Midkine as a Biomarker in Ovarian Cancer

Midkine was identified as a potential biomarker for differentiating ovarian cancer from benign tumors and population controls. In distinguishing benign tumors from ovarian cancer, a marker panel that included MDK was found to have a significantly higher area under the curve (AUC) than either cancer antigen 125 (CA125) or human epididymis protein 4 (HE4), which are the standard biomarkers for ovarian cancer [146]. This panel had higher sensitivity but lower specificity than OVERA^®^ and improved upon both the sensitivity and specificity of either CA125 or HE4 in detecting stage III–IV ovarian cancer [146]. MDK is significantly elevated in ovarian cancer relative to normal tissues, and MDK levels could detect ovarian cancer samples from normal tissues and benign tumors [147]. However, when combined with CA125 and HE4, MDK did not significantly increase the predictive capacity [147]. Similarly, Rice et al. demonstrate that while MDK does have significant independent predictive ability for ovarian cancer, it is less sensitive than CA125. However, combination with CA125 and anterior gradient protein 2 homolog (AGR2) resulted in a panel with significantly better predictive ability than CA125 alone [148]. Another study found that serum MDK demonstrated a significant increase from healthy patients to late-stage ovarian cancer, with benign and early-stage tumors serving as intermediate points [149]. Of the proteins assessed, MDK had the third-highest sensitivity when measured at a specificity of 95%. In a receiving operating characteristic curve for separating ovarian cancer from benign tumors, MDK was the fourth best. Notably, MDK did have some weaknesses, as it was less helpful at discriminating early-stage ovarian cancer from benign disease and had poor sensitivity in the cancer vs. benign comparison groups [149]. Taken as a whole, these data suggest that while MDK may not be sufficient as a solo biomarker, it has abundant potential as a component of ovarian cancer biomarker panels.

### 4.5. Midkine in Endometrial Cancer

MDK is frequently overexpressed in endometrial cancer [150,151,152]. Similarly, endometrial cancer progression has been demonstrated to be correlated with the expression of known MDK receptors such as ALK [153] and Notch2 [154]. Overexpression of ALK is associated with higher tumor grade, enhanced stemness, EMT, migration, and proliferation, as well as decreased apoptosis, in endometrial cancer [153,155]. Single-cell RNA sequencing has indicated that MDK may be able to mediate the communication of a malignant phenotype from endometrial cancer cells to endothelial cells via an MDK-NCL signal [48]. Furthermore, this data suggests that MDK may shape the tumor microenvironment by inhibiting immune cells and modulating stromal cells via MDK-NCL [48]. However, the results of this sequencing study remain to be tested experimentally. MDK inhibition via imatinib mesylate, a tyrosine kinase inhibitor, has been proposed as a mechanism by which the drug reduces endometrial cancer viability, with the addition of medroxyprogesterone acetate contributing to a further drop in both MDK and viability [156]. This reduction of MDK in response to a progesterone treatment further supports a potential role for MDK as an estrogen-related gene and suggests a potential role for MDK in estrogen-driven women’s cancers such as endometrial cancer—a role that remains to be studied (Figure 3).

### 4.6. Midkine as a Biomarker in Endometrial Cancer

The potential of MDK as a biomarker in endometrial cancer is a topic of debate. However, MDK is frequently overexpressed in endometrial cancer tissues. One study found that MDK was elevated in the serum of endometrial cancer patients relative to benign gynecological tumors, and tumors had higher MDK expression relative to healthy tissue [152]. Furthermore, preoperative serum concentrations of MDK correlated with both the lymph node metastasis status and the prognosis of endometrial cancer patients [152]. A subsequent study identified association between MDK and endometrial cancer, suggesting it could be included in a diagnostic panel for detection [146]. However, its utility as a standalone biomarker was not supported. Contradicting other findings, one study reported no elevation of MDK in endometrial cancer, instead observing increased levels in patients with endometriosis [157]. This aligns with previous evidence of elevated MDK in endometriosis [64]. This highlights a potential challenge in using MDK as a biomarker in endometrial cancer, as upwards of 10% of reproductive-age women have endometriosis [158].

### 4.7. Midkine in Cervical Cancer

Of the women’s cancers, cervical cancer is the one in which MDK has been studied the least. Nevertheless, some evidence exists that MDK is elevated in cervical cancers [150]. Fei et al. have demonstrated that MDK suppresses apoptosis via apoptosis regulator BAX reduction and BCL2 increase, and promotes colony formation, invasion, and migration in vitro. Furthermore, MDK promotes tumor formation and growth, and increases the number of lymph node metastases [159] (Figure 3). MDK also affects the tumor microenvironment, promoting lymphangiogenesis and impairing lymphatic barrier function [159]. MDK has been indirectly implicated in promoting proliferation, growth, invasion, migration capacity, and chemotherapy resistance via tumor-associated calcium signal transducer 2 (TROP-2) [160]. Mechanistic studies have shown that TROP-2 acts to suppress the aforementioned processes via competitively binding MDK and decreasing the MDK-mediated activation of ALK. However, this study was limited by the fact that the functional consequences of MDK were never directly assessed. Rather, it was inferred through the fact that TROP-2 binds MDK, and that inhibiting ALK mitigated the effects of TROP-2 overexpression [160]. Some of these putative roles for MDK have been confirmed in other studies, though. In cervical cancer cells, MDK was capable of promoting metastasis, migration, and invasion via EMT in response to IFN-γ treatment via STAT1 [109]. Supporting a role for MDK in promoting drug resistance, another study found that MDK overexpression in HeLa cells promoted drug resistance to 5-fluorouracil (5-FU), doxorubicin, and cisplatin, although no mechanism was explored in this study [161]. Another putative role for MDK is mediating communication within the tumor and its microenvironment. MDK seems to be involved in tumor cell communication through binding several of its receptors, including syndecan 1 (SDC1), SDC4, NCL, and LRP1, and MDK from tumors seems to bind to NCL on macrophages and lymphocytes, though the functional consequences have not been explored [162]. Further evidence indicates that MDK from cervical cancer stem cells promotes M2 polarization via MDK-LRP1 and MDK-SORL1 interactions with macrophages [163].

### 4.8. Midkine as a Biomarker in Cervical Cancer

Although there is evidence that MDK is elevated in cervical cancer, there are few studies that have conducted further research on this topic. The expression of MDK mRNA and proteins have been found to be elevated in cervical cancer tissues relative to non-cancerous cervical tissues, with protein expression also correlating with tumor size and stage [164]. The serum level of MDK was also higher in cervical cancer patients than in healthy control subjects, and it correlated with stage, tumor size, and muscle invasion, but not with para-uterine invasion, age, or vascular invasion [159]. Notably, MDK in cervical cancer patients with lymph node metastases was higher than in patients without lymph node metastasis, with a significant AUC and moderate sensitivity and specificity. When combined with squamous cell carcinoma antigen (SCCA), the current cervical cancer biomarker in use to predict lymph node metastasis and prognosis, the combination of SCCA and MDK had a higher AUC than either alone, indicating higher diagnostic accuracy [159]. Thus, future studies into the utility of MDK as a prognostic factor may be helpful.

## 5. Therapeutic Targeting of MDK

Due to MDK’s diverse effects on cancer cells and the fact that MDK-deficient mice exhibit no major abnormalities, MDK makes an appealing target in women’s cancers [165]. To date, a handful of strategies have been used to target MDK, including MDK-binding peptides [94], antisense oligonucleotides [92], and most notably, a small molecule inhibitor of MDK [166]. This small molecule inhibitor, iMDK (full name 3-[2-(4-fluorobenzyl) imidazo [2,1-beta] thiazol-6-yl]-2*H*-chromen-2-one), functions by suppressing endogenous MDK expression through an unspecified mechanism [166]. In the years since iMDK was initially reported, a number of preclinical studies have made use of it, demonstrating promising preclinical anti-cancer activity including growth suppression, enhancement of apoptosis, and inhibition of migration across a variety of cancers [87,95,166]. Furthermore, iMDK has exhibited utility in reducing the number of IFN-γ-induced metastases across numerous cancers, including breast and cervical cancers [109] (Figure 4).

These results using iMDK in various cancers are interesting; however, there are some notable limitations. On the one hand, iMDK has shown potential off-target activity. Co-treatment of MDK and a MEK inhibitor displayed combination effects in a cell line without MDK [167]. iMDK has also demonstrated an ability to kill primary effusion lymphoma cells, despite their lack of MDK expression [168]. In this case, MDK instead caused an inhibition of CDK-activating kinase (CAK) [168]. As CAK is a regulator of the cell cycle, this raises a concern for generalized toxicity. Another possible weakness is that iMDK only lowers the expression of MDK and does not act directly on MDK [167]. Furthermore, the iMDK inhibitor identified through a promoter–reporter assay represents an indirect target, with its actual target being one of the regulators that modulate the MDK promoter. Tumor cells can quickly develop resistance to such inhibitors by developing alternative means to express MDK. Therefore, methods of inhibiting MDK directly may have more clinical utility. Treatment with functional antibodies against MDK suppressed growth of osteosarcoma cell lines [169]. Considering the significance of MDK and its potential as an anti-cancer target, a small biotech company is developing MDK antibodies for clinical use. Currently they are in early stages of development (https://www.roquefortplc.com/). Our laboratory recently identified HBS-101 as a potent MDK inhibitor (Figure 3 and Figure 4). Molecular docking studies showed that HBS-101 interacts at the MDK binding interface and microscale thermophoresis studies confirmed the direct interaction of HBS-101 with MDK. Treatment with HBS-101 significantly reduced the proliferation of TNBC cells both in vitro and in vivo. The HBS-101 is currently in preclinical evaluation and may represent a first-in-class MDK inhibitor [170].

A number of studies have suggested that the inhibition of MDK can enhance immunotherapy. Single-cell analysis has implicated MDK in reducing the reinvigoration of T cells following PD-1 inhibition [124]. In hepatocellular carcinoma, MDK overexpression decreased the expression of programmed death ligand 1 (PD-L1) in MDSCs, promoting immunosuppression and reducing the cytotoxicity of T cells. When tested in vivo, MDK inhibition enhanced the efficacy of sorafenib and a PD-L1 inhibitor [103]. Cerezo-Wallis et al. demonstrated a significant combination effect when mice were treated with iMDK and anti-PD-L1 therapy. They identified MDK as a key player in facilitating tumor-promoting NF-κB signaling, macrophage recruitment, and M2 polarization, as well as suppressing a tumor-suppressive interferon (IFN) response [112].

Besides iMDK, BO-110, a double-stranded RNA nanoplex which functions through repression of *MDK* and *VEGFR3* gene transcription via IFN-α/β, has recently been investigated as a possible therapeutic [171]. This drug was capable of limiting metastasis and neolymphangiogenesis in melanomas [171]. While there are numerous studies investigating iMDK across cancers, the women’s cancers remain largely neglected in this regard. There is one study indicating utility of iMDK in enhancing IFN-γ therapy in ovarian cancer [141]. In this study, iMDK was able to enhance the growth suppression caused by IFN-γ therapy while also reducing the EMT and metastasis that IFN-γ promoted [141]. Aside from this and the aforementioned evidence that iMDK reduces IFN-γ-mediated EMT in breast and cervical cancers [109], the utility of MDK small molecule inhibitors has not been interrogated. As such, future studies examining the utility of MDK inhibition in women’s cancers would be very beneficial.

## 6. Conclusions and Future Directions

In summary, these emerging studies suggest that MDK signaling contributes to the progression of women’s cancers. By interacting with cell surface receptors and then activating the PI3K/AKT, STAT3, and MAPK/ERK pathways, MDK coordinates a series of signaling events that promote cancer cell stemness and cell proliferation. MDK overexpression frequently occurs in women’s cancers and is linked to aggressive tumor behavior, treatment resistance, and metastasis. Additionally, MDK signaling plays a role in control of immune cell recruitment, cytokine synthesis, and angiogenesis. Collectively, these findings establish MDK as a potential therapeutic target and biomarker. However, the published studies employed a limited number of tumor samples that limit MDK’s potential as a biomarker. Further research utilizing an increased number of tumor samples is evidently required to establish MDK’s utility as a biomarker either alone or in combination with other established biomarkers. In preclinical investigations, the progression of numerous cancers has been impeded by the inhibition of MDK signaling through genetic knockdown or indirect inhibition of receptor signaling. These studies indicate that combination therapies that incorporate MDK signaling in conjunction with critical receptor signaling antagonists or immune check point inhibitors are ideal for better therapeutic strategies. The lack of small molecule inhibitors that directly bind and block MDK signaling represents a major knowledge gap and a critical barrier to the advancement of MDK therapy. Given that MDK is a soluble secreted protein, the development of antibody-based treatments is a viable therapeutic option. Nevertheless, antibody-directed therapy for MDK inhibition is expensive, necessitating the development of more affordable alternatives, such as small molecule inhibitors targeting MDK to provide optimal patient care and potentially save millions of dollars on healthcare costs. As such, small molecule inhibitors targeting MDK may have utility in treating women’s cancers.

## Figures and Tables

**Figure 1 ijms-26-04809-f001:**
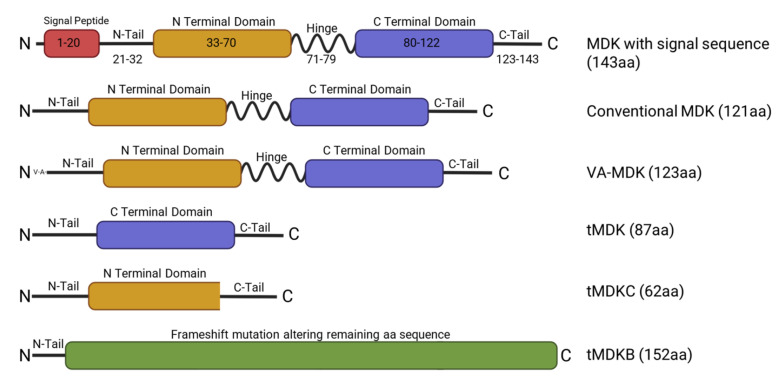
Schematic representing MDK full length and isoforms and their respective structural domains. The MDK protein chain (aa 21–143) is made of two domains (the N domain and the C domain). Both domains are linked together by disulfide linkages. The full length MDK has a molecular weight of 15.5 kilodaltons and contains a signal peptide for secretion (aa 1–20). The C-terminal domain is important for MDK action, and the N-terminal domain is necessary for dimerization. tMDK/tMDKC/tMDKB, MDK isoforms encoded by truncated MDK mRNA transcripts; VA-MDK, valine-alanine MDK.

**Figure 2 ijms-26-04809-f002:**
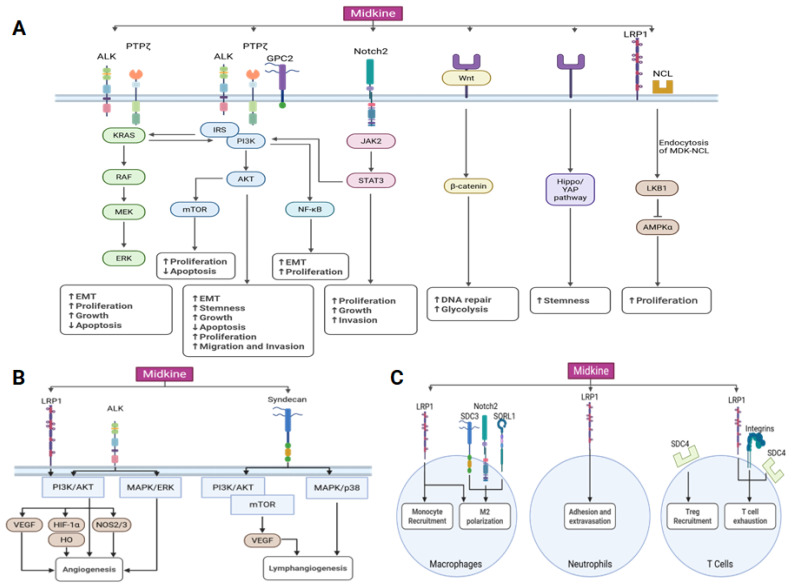
MDK oncogenic signaling network. (**A**) MDK signaling facilitates tumor growth by modulating many signaling pathways, including the activation of STAT3, ERK, AKT, MAPK, β-catenin, Hippo/Yap, and AMPKα. The MDK signaling axis facilitates cell survival, augments stemness, and suppresses apoptosis. The upward arrow (↑) denotes an increase, while the downward arrow (↓) signifies a decrease in signaling pathway activity. (**B**) MDK signaling also affects angiogenesis and lymphangiogenesis. (**C**) MDK signaling also influences tumor microenvironments by regulating the functions of macrophages, neutrophils, and T lymphocytes. ALK, anaplastic lymphoma kinase; AMPK, AMP-activated protein kinase; GPC2, glypican-2; HO, heme oxidase; IRS, insulin receptor substrate; LRP1, low-density lipoprotein receptor-related protein 1; NCL, nucleolin; PTPζ, protein tyrosine phosphatase ζ; SORL1, sortilin-related receptor 1; SDC3, syndecan 3; SDC4, syndecan 4.

**Figure 3 ijms-26-04809-f003:**
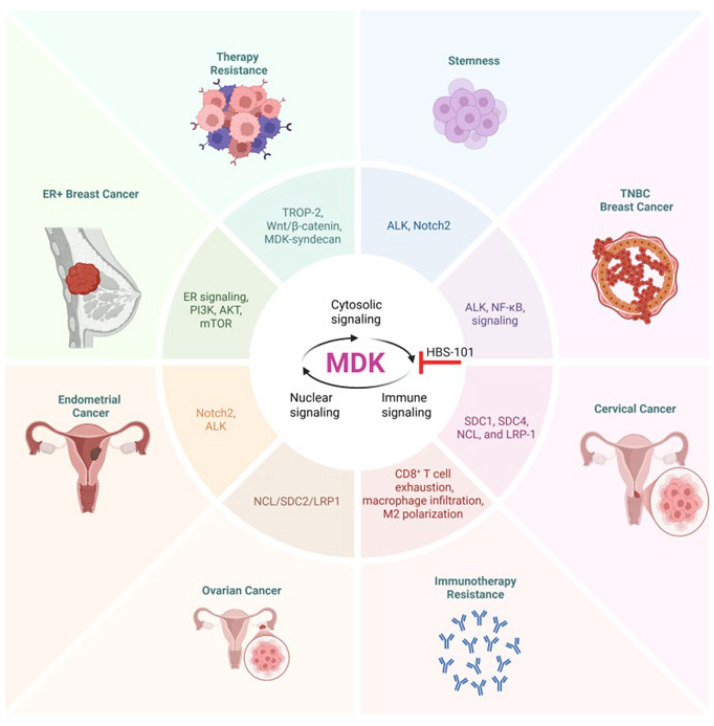
Significance of MDK signaling in women’s cancers. In ER-positive breast cancer, the course of the disease is promoted by MDK signaling via PI3K, AKT, and mTOR. MDK signaling through ALK and NF-kB is thought to contribute to the progression of TNBC. Some of the factors that contribute to the advancement of endometrial cancer include MDK signaling through Notch and ALK receptors. Ovarian cancer progression is influenced by MDK signaling through NCL/SDC2/LRP1, which plays a role in development. There is also evidence that MDK signaling contributes to stemness, chemoresistance, and resistance to immunotherapy in women’s cancers. ALK, anaplastic lymphoma kinase; ER, estrogen receptor; LRP1, low-density lipoprotein receptor-related protein 1; NCL, nucleolin; SDC1/2/4, syndecan 1/2/4; TROP2, trophoblast cell surface antigen 2.

**Figure 4 ijms-26-04809-f004:**
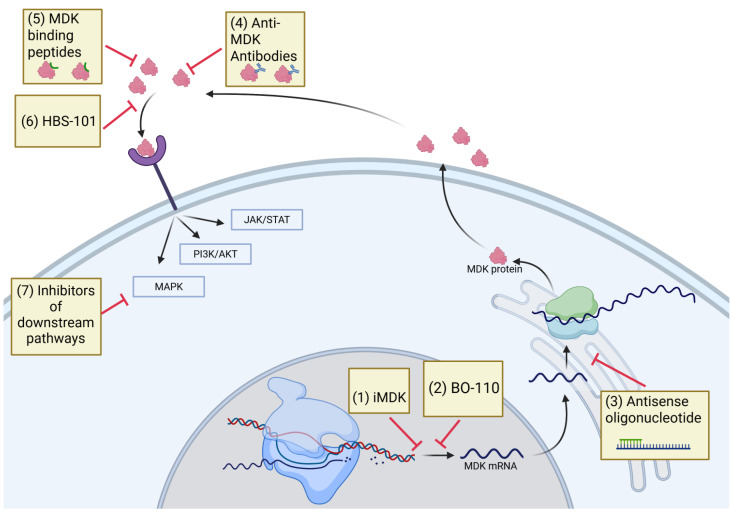
Schematic illustrating the pharmacological agents that obstruct MDK downstream signaling. The anti-MDK antibody iMDK and the MDK inhibitor HBS-101 can disrupt MDK signaling. Inhibiting MDK signaling through the use of inhibitors that target MDK-activated pathways, such as JAK/STAT, PI3K, and MAPK inhibitors, may also effectively disrupt MDK signaling. JAK/STAT, Janus kinase/signal transducers and activators of transcription; PI3K, phosphatidylinositol 3-kinase; MAPK, mitogen-activated protein kinase.

## Data Availability

Not applicable.

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
