# Peer review of "Significance of Midkine Signaling in Women’s Cancers: Novel Biomarker and Therapeutic Target"

_ijms, 2025, doi:10.3390/ijms26104809_

Round 1
Reviewer 1 Report
Comments and Suggestions for Authors
- Page 2, Midkine structure and regulation: The reviewer believes that the detailed structure of the MDK should be shown in the figures, and the authors have to create this figure in this section. For example, it should show about seven different isoforms of mature proteins or different aspects of the structures of VA-MDK and tMDK. In addition, the process of formation of the seven different isoforms of MDK should be shown.
- Page 7, line251 and page 8, line294: A part of these sentences is underlined: What does it mean? Or is it a mistake?
- Page 10, table 1: The reviewer believes that it is essential to demonstrate the function of MDK induced by receptors. However, the reviewer could not understand these functions in this table. The reviewer suggests that the authors explain the functions in more detail and review the writing methods
Reviewer 2 Report
Comments and Suggestions for Authors
This manuscript is a well-structured review of the importance of Midkine in women's cancers. Although the information presented in the text is very good, the review could be improved by adding more figures that summarize the information in the text, for example, a figure highlighting the findings on the functional role of Midkine only in gynecological cancers, and a figure schematizing the secretion and function of this molecule, according to the authors' criteria.
Revisions:
- "3.2. Midkine in cell proloferation", line 107, "proliferation"
- Insert Figure 1 after the last paragraph that describes it.
- The format of Table 1 could be modified and placed immediately after the paragraph that cites it.
- Sections 4 and 5 may be somewhat repetitive. Perhaps the information on Midkine signaling and potential biomarkers in women's cancers could be integrated into a single section. Why did the authors feel this information should be separate?
- Lines 445-446; The statement seems contradictory; MDK levels are not associated with a worse prognosis, but they are associated with metastasis... ?
- In the Statements section: Data Availability Statement, Acknowledgments and Conflicts of Interest, the authors did not add the corresponding information, the manuscript shows the indications described in the journal template.
- In some sections, the manuscript has too many acronyms or abbreviations to make it difficult for a non-expert reader to read the text fluently. It would be wise to leave only the necessary abbreviations in the manuscript as a whole.
Round 2
Reviewer 1 Report
Comments and Suggestions for Authors
The revised paper seems to have improved and is worth publishing in this journal.
